# Reversible Tumor Progression Induced by a Dexamethasone Course for Severe COVID-19 during Immune Checkpoint Inhibitor Treatment

**DOI:** 10.3390/diagnostics12081933

**Published:** 2022-08-10

**Authors:** Paul Gougis, Baptiste Abbar, Julie Benzimra, Aurore Vozy, Jean-Philippe Spano, Luca Campedel

**Affiliations:** 1Sorbonne Université, INSERM, Institut Pierre Louis d’Epidémiologie et de Santé Publique (iPLESP), Assistance Publique-Hôpitaux de Paris (AP-HP), Pitié Salpêtrière Hospital, Department of Medical Oncology, Institut Universitaire de Cancérologie, CLIP^2^ Galilée, 75013 Paris, France; 2Residual Tumor & Response to Treatment Laboratory, RT2Lab, INSERM, U932 Immunity and Cancer, Institut Curie, 75005 Paris, France; 3Sorbonne Université, Department of Radiology, Assistance Publique-Hôpitaux de Paris (AP-HP), Pitié Salpêtrière Hospital, 75004 Paris, France; 4Service d’Oncologie Médicale, CHU Gabriel Montpied, Université Clermont Auvergne, 63000 Clermont-Ferrand, France

**Keywords:** immunotherapy, immune checkpoint inhibitors, COVID-19, pseudoprogression, corticosteroids

## Abstract

Immunotherapies and immune checkpoint inhibitors (ICI) represent the latest revolution in oncology. Several studies have reported an association between the use of corticosteroids and poorer outcomes for patients treated with ICIs. However, it has been never established whether corticoid-induced tumor progression under ICI treatment could be reversible. We report herein transient tumor progression induced by dexamethasone for a patient treated with pembrolizumab for metastatic bladder cancer. An 82-year-old man was treated with pembrolizumab as a second-line treatment for metastatic urothelial carcinoma with stable disease for 8 months as the best tumoral response. He experienced severe coronavirus disease 2019 (COVID-19) infection and was treated with high-dose dexamethasone for ten days according to the RECOVERY protocol. Following this episode, radiological CT-scan evaluation showed tumor progression. Pembrolizumab was maintained, and subsequent radiological evaluation showed tumor shrinkage. This case highlights that the antagonistic effect of glucocorticoids with ICI efficacy is transient and can be reverted when corticoids are withdrawn. Clinicians should be aware that tumor progression in the context of the intercurrent use of systemic corticosteroids can be temporary and should be interpreted with caution, and ICI continuation could be considered for some patients. **Insights:** The antagonistic effect of glucocorticoids with ICI efficacy is transient and can be reverted when corticoids are withdrawn. Tumor progression in the context of the intercurrent use of systemic corticosteroids can be temporary and should be interpreted with caution, and ICI continuation could be considered for some patients.

**Figure 1 diagnostics-12-01933-f001:**
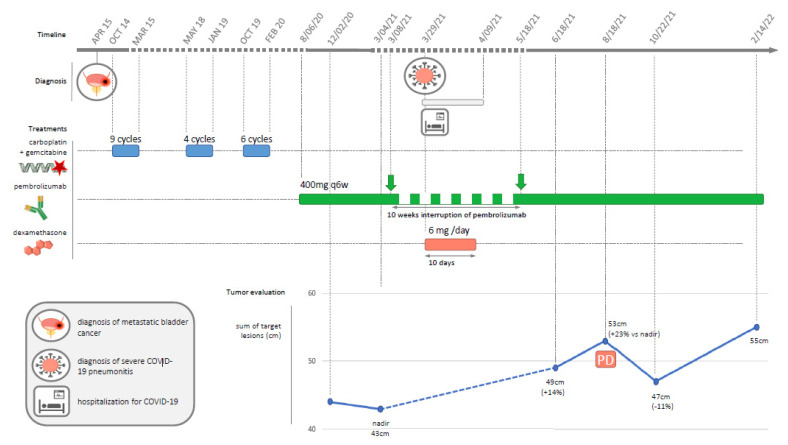
Timeline of patient history depicting cancer diagnosis, cancer treatments, and COVID-19 diagnosis and treatment. In April 2014, a 75-year-old man with a history of smoking (20 pack-years) presented with hematuria, which revealed an FGFR wild-type urothelial bladder carcinoma with lung metastases. The patient was also treated for hypertension and diabetes. He was treated with a combination of gemcitabine–carboplatin as a first-line treatment between October 2014 and March 2015 (n = 9 cycles). After a partial response (RECIST 1.1 criteria) was observed on CT-scan evaluation, he was monitored without treatment. He relapsed in May 2018 and was rechallenged with gemcitabine–carboplatin (n = 4 cycles) until January 2019, when he achieved another partial response. For the third relapse in October 2019, he was treated with six more cycles of gemcitabine–carboplatin until February 2020, with stable disease as the best response. Tumor progression was diagnosed in July 2020, and pembrolizumab at the dose of 400 mg (intravenous every 6 weeks) was initiated on 6 August 2020, according to local guidelines. The patient had stable disease as the initial best response. After eight months of stable disease with pembrolizumab, the patient experienced severe coronavirus disease 2019 (COVID-19) infection and was admitted to the intensive care unit on 29 March 2021. He was treated with oxygen (up to 12 L/min without noninvasive ventilation) and 6 mg of dexamethasone per day for ten days according to the RECOVERY protocol [1]. The clinical evolution was favorable, with oxygen withdrawal and hospital discharge on 9 April 2021. Pembrolizumab was resumed on 18 May 2021 after an interruption of ten weeks. Radiological CT-scan evaluation on 18 August 2021 showed RECIST tumor progression (Figure) with a target lesion sum of 53 mm compared to 43 mm (+23%) on nadir (4 March 2021). Pembrolizumab injections were maintained, and the following radiological evaluation on 22 October 2021 showed a decrease of 11% (sum of target lesion = 47 mm vs. 53 mm, stable disease according to RECIST criteria) compared to August 2021 (Figure).

**Figure 2 diagnostics-12-01933-f002:**
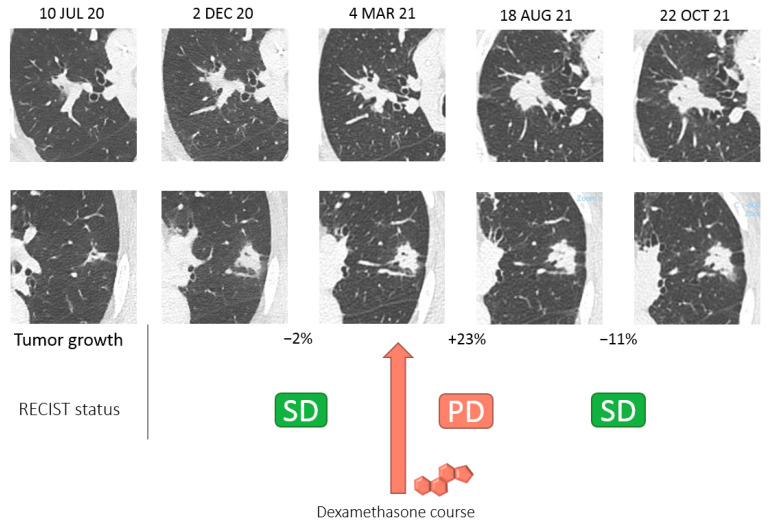
Timeline of TDM scan evaluation of the two target lesions according to RECIST 1.1 criteria. Glucocorticoids (GC) are among the most potent anti-inflammatory and immunosuppressive agents. In severe COVID-19 infection, the host immune response is thought to play a key role in the pathophysiology of organ failure, and the use of dexamethasone to mitigate inflammatory organs has been shown to result in lower mortality [2,3]. Immune checkpoint inhibitors (ICI) such as pembrolizumab have been successfully developed for the treatment of various cancers and have been approved for several indications, including for bladder cancer. They potently restore exhausted T-cell functions and are activated at tumor contact [4]. Corticoids have multiple mechanisms and inhibit several lymphocyte activities, such as T cell differentiation, activation, and migration [5]. Therefore, there is a strong rationale for pharmacological antagonism between corticoids and ICI. GCs could impair the antitumoral activity of ICI, and this negative pharmacological interaction also explains their efficacy as a treatment for the immune-related adverse events induced by ICI [6]. Several retrospective studies have reported a negative impact of the use of corticoids on the outcomes of patients treated with ICI [7,8,9,10], and this association was further confirmed in a meta-analysis of 4045 cancer patients [11]. Based on these findings, the EMA recommends avoiding the use of systemic corticosteroids before starting pembrolizumab [12]. However, despite efforts made to limit confusion biases in these studies, the causality assessment of the negative impact of corticoids remains limited. In this case report, we demonstrate for the first time that the antitumoral effect of pembrolizumab can be temporarily antagonized by a strong course of GCs but that the antitumoral effect is later recovered and translated into tumor shrinkage and control. Of note, pembrolizumab cessation for 4 weeks compared to the classic scheme (10 weeks instead of 6 weeks) could not explain tumor progression when taking the long-lasting effects of a pembrolizumab concentration several times above the inhibitory concentration 50 and treatment half-life (3 weeks) into consideration [4]. Dexamethasone has a short half-life of fewer than three days, and no active concentration remains after two weeks. In our case, however, we observed tumor progression between 40 and 100 days after dexamethasone cessation followed by tumor decrease. The direct antagonistic effect of GCs could not explain such a chronology. However, GCs have a pleiotropic impact on different T cell populations and precursors and can regulate T cell trafficking, alter TCR repertoire, or modify thymic homeostasis, which could explain the long-lasting and remnant antagonizing effects despite a short half-life [13]. Pseudoprogression could be an alternative explanation for transient tumor progression, and is not an exceptional event with urothelial cancer, with 1.5% to 17% of cases being treated with PD-1 inhibitors [14]. However, pseudoprogression arises exceptionally after three months (late pseudoprogression), and no cases have been reported after 6 months of treatment for urothelial cancer or other tumor types [15,16]. Classic pseudoprogression seems unlikely in this case, with one year of treatment before transient progression was observed. However, we could not discard the late pseudoprogression induced by GC cessation followed by transient inflammation that led to tumor progression followed by a decrease in tumor size. Consequently, tumor progression in the context of the intercurrent use of systemic corticosteroids should be interpreted with caution due to potential pharmacological antagonism, and ICI continuation should be considered for some patients. Tumor evaluation was carried out according to RECIST 1.1 criteria. COVID-19: coronavirus disease 2019; PD: progressive disease (RECIST 1.1 criteria). The tumor progressed after a course of dexamethasone and later experienced tumor shrinkage that did not meet partial response according to RECIST 1.1 criteria. PD: progressive disease; SD stable disease.

## Data Availability

Not applicable.

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
