# Peer review of "Reversible Tumor Progression Induced by a Dexamethasone Course for Severe COVID-19 during Immune Checkpoint Inhibitor Treatment"

_diagnostics, 2022, doi:10.3390/diagnostics12081933_

Round 1

Reviewer 1 Report

The manuscript entitled “Reversible tumor progression induced by a dexamethasone course for severe COVID-19 during immune checkpoint inhibitor treatment - a case report” describes a patient with metastatic bladder cancer under treatment with pembrolizumab, who experienced a reversible disease progression during dexamethasone treatment.

The observed phenomenon is interesting in that the antagonistic effect of steroid against pembrolizumab treatment was reversible and the tumor shrinkage (stable disease) was observed after cessation of steroid.

However, there is room for improvement.

Major:

I think that the pseudoprogression should be fully considered during the differential diagnosis.

I agree with the authors in that a cessation of pembrolizumab could not explain the disease progression. However, we sometimes experience pseudoprogression during the continuation of immune checkpoint inhibitors, which could occur several times.

Therefore, the described clinical course and discussion would not be enough to conclude that the observed phenomenon was due to antagonistic effect of steroid.

This point is the most important point, and I recommend the author to elucidate the reason why the pseudoprogression could be excluded.

Minor:

1)     The images of lung metastases are not typical, which could be due to the influence of prior treatments. I recommend the authors to add the pretreatment image in Figure 2.

2)     There are some grammatical errors, and the manuscript should be reviewed by a native speaker.

Reviewer 2 Report

The current manuscript discusses the pharmacological antagonism between Immune check inhibitors that elicit the immune response against cancer and dexamethasone (immunosuppressant) that helps manage the cytokine storm of COVID-19 in an elderly patient.

The case is quite interesting and shows how COVID-19 - cancer interacts. It could be publishable work and prone to be citable in oncology rather than the COVID-19 field. 

Optional Suggestion to improve the manuscript:

- The manuscript could be more organized to clarify the pharmacological antagonism (pharmacodynamic antagonism) and easy to flow.

- Also, it would be great to introduce the term pharmacological antagonism in the text and one or two sentences to describe the pharmacodynamic antagonism

- The title should be shorten and sharp in a way that doesn't affect the meaning. 

- The following manuscript might be useful in citation: Pathogenesis and Management of COVID-19, Alfarouk et al., Xenobiotics, 2021

Round 2

Reviewer 1 Report

The manuscript has been well revised.

However, I am uncertain whether the observed phenomenon could be solely explained by the reversible progressive disease caused by systemic steroid treatment for COVID-19.

Dexamethasone had been administered until Apr. 9 and pembrolizumab was resumed on May 18, which was followed by the RECIST-PD on Aug. 18.

The duration of antagonistic effect of dexamethasone seems too long.

And there are also several reports on the cases which showed durable response of ICI during its cessation as well as the steroid treatments for irAEs. The author should consider these reports and clarify the difference between the current case and them. The influence of COVID-19 on the immune system and the efficacy of ICI could be one explanation.

Finally, the additional references (Refs. 13-15) are not enough to consolidate the scientific basis for excluding pseudoprogression. The reports on pseudoprogression after durable response tend to be underestimated, because the interval between CT evaluations tends to be longer under durable response.

Therefore, I recommend the authors to revise the manuscript to elucidate the above-mentioned points.

Author Response

The manuscript has been well revised.

However, I am uncertain whether the observed phenomenon could be solely explained by the reversible progressive disease caused by systemic steroid treatment for COVID-19.

Dexamethasone had been administered until Apr. 9 and pembrolizumab was resumed on May 18, which was followed by the RECIST-PD on Aug. 18.
The duration of an
tagonistic effect of dexamethasone seems too long.

 Dexamethasone has a short half-life of about 2 to 3 days. Within 15 days, dexamethasone concentrations cannot remain active. In our case report, we observed a tumor keeping its progression course 6 weeks after dexamethasone interruption, raising interesting questions regarding the mechanism by which dexamethasone could have impacted the tumor course.

The best-known effect of glucocorticoids is their powerful ability to suppress the immune response. However, dexamethasone has a pleiotropic impact on different T cell populations [Taves, Nat. Rev. Immunol., 2021]
Glucocorticoids could:
- regulate T cell trafficking, resulting in decreased effector and memory responses
- alter TCR repertoire by antagonizing negative selection
- modify thymic homeostasis with thymic involution closely and dynamically, following changes in systemic GC levels

 Therefore, from a mechanistic point of view, we cannot assume that the antagonizing effect of glucocorticoids could not be remnant and long-lasting despite the short half-life of dexamethasone.

 Finally, we could not totally discard a late pseudoprogression. Perhaps, glucocorticoids cessation was followed by transient inflammation that led to tumor progression followed by tumor size decreased, although no element from the literature can support this hypothesis.

 We implemented these important considerations to the manuscript.

And there are also several reports on the cases which showed durable response of ICI during its cessation as well as the steroid treatments for irAEs. The author should consider these reports and clarify the difference between the current case and them. The influence of COVID-19 on the immune system and the efficacy of ICI could be one explanation.

 Pasola and colleagues [Pesola, Immunotherapy, 2022] described a case of a metastatic RCC with a long response after corticoids and treatment cessation. In this case, corticoids were introduced for severe immune-related adverse events (irAE). A deep tumor response could have happened earlier which would explain a late disease-free status.

In our case, there was no important tumor responses. The balance between tumor control and tumor progression was probably tighter, which could explain why corticoids would tip the balance in favor of tumor progression and mild tumor decrease secondly.

 Patients experiencing severe irAEs have better tumor responses with or without corticoids with different tumor types [Foster Cancer 2021; Sung J Thorac Dis. 2020]. In our case, the patient was treated with a corticoid course for an infectious reason and not an irAE, which could explain that the tumor behavior was different than what was previously described with irAEs.

Finally, the additional references (Refs. 13-15) are not enough to consolidate the scientific basis for excluding pseudoprogression. The reports on pseudoprogression after durable response tend to be underestimated, because the interval between CT evaluations tends to be longer under durable response.

 We addressed this question above and modified the manuscript consequently.

Thank you for your consideration,

Round 3

Reviewer 1 Report

The manuscript has been well revised and become more scientifically sound.

I think that no one can exclude the possibility of pseudoprogression, and the recognition of uncertainty in the current case as well as the scientific speculation.

The observed phenomenon could be explained by the delayed antagonistic effects of steroid, and I agree with the authors’ discussion.

Minor:

1)     There are some grammatical errors in the revised version, which should be checked by a native speaker.

Author Response

The manuscript has been well revised and become more scientifically sound.
I think that no one can exclude the possibility of pseudoprogression, and the recognition of
uncertainty in the current case as well as the scientific speculation.

The observed phenomenon could be explained by the delayed antagonistic effects of steroid, and I
agree with the authors’ discussion.

Minor:

1) There are some grammatical errors in the revised version, which should be checked by a
native speaker.

We revised our manuscript with an English native speaker

Thank you for your consideration
